# Immunoglobulin G N-glycan Biomarkers for Autoimmune Diseases: Current State and a Glycoinformatics Perspective

**DOI:** 10.3390/ijms23095180

**Published:** 2022-05-06

**Authors:** Konstantinos Flevaris, Cleo Kontoravdi

**Affiliations:** Department of Chemical Engineering, Imperial College London, London SW7 2AZ, UK

**Keywords:** autoimmune disorders, glycosylation, glycoinformatics, artificial intelligence, systems biology, precision medicine

## Abstract

The effective treatment of autoimmune disorders can greatly benefit from disease-specific biomarkers that are functionally involved in immune system regulation and can be collected through minimally invasive procedures. In this regard, human serum IgG N-glycans are promising for uncovering disease predisposition and monitoring progression, and for the identification of specific molecular targets for advanced therapies. In particular, the IgG N-glycome in diseased tissues is considered to be disease-dependent; thus, specific glycan structures may be involved in the pathophysiology of autoimmune diseases. This study provides a critical overview of the literature on human IgG N-glycomics, with a focus on the identification of disease-specific glycan alterations. In order to expedite the establishment of clinically-relevant N-glycan biomarkers, the employment of advanced computational tools for the interpretation of clinical data and their relationship with the underlying molecular mechanisms may be critical. Glycoinformatics tools, including artificial intelligence and systems glycobiology approaches, are reviewed for their potential to provide insight into patient stratification and disease etiology. Challenges in the integration of such glycoinformatics approaches in N-glycan biomarker research are critically discussed.

## 1. Introduction

The discrimination between self-antigens and non-self-antigens lies at the core of immunology and is imperative for well-regulated innate and adaptive immunity. The aberrant adaptive immune response targeting self-antigens—also referred to as “autoantigens”—is termed “autoimmunity”, and the associated disorders are called autoimmune diseases [1]. The United States Autoimmune Association (https://autoimmune.org/ (accessed on 17 February 2022)) has reported the existence of more than 100 autoimmune diseases collectively affecting approximately 4–5% of the world’s population [2]. The etiology of these diseases, characterised by both genetic predisposition and environmental triggers, has yet to be completely elucidated, and effective treatment hinges on timely diagnosis and monitoring [1,3,4]. However, the identification of the onset of an autoimmune disease in a patient is hampered by the fact that there is no single highly specific diagnostic test that can confirm the presence of a particular autoimmune disorder; rather, multiple laboratory tests are needed, including a complete blood count, serologies, cytokine analysis, and acute phase reactants [3]. Furthermore, in multiple sclerosis (MS), which is a neurodegenerative autoimmune disease, cerebrospinal fluid (CSF) is routinely collected by lumbar puncture [5], which, although safe, is an invasive procedure that is uncomfortable for the patient. Thus, the diagnosis and monitoring of autoimmune diseases would greatly benefit from disease-specific biomarkers, preferably collected by minimally invasive means, which could be used to identify the disease development compared to healthy controls, stratify patients based on disease severity, and quantify the patient response to therapy [6,7].

The emergent field of glycomics holds great promise for the advancement of biomarker research in the context of autoimmune diseases, with the goals to uncover disease predisposition and development, and to assist in the identification of specific molecular targets for advanced therapies [8,9,10,11]. Glycomics refer to the study of the glycome, which constitutes the entire set of glycans present in an organism, tissue, cell, or protein [12]. Glycans are natural biopolymers that are highly diverse in structure, with profound biological and immunological significance [13,14,15], which are known to be explicitly involved in every major disease pathophysiology [16]. Their functional role in the latter remains to be fully characterised, partly due to the complexity of glycan biosynthesis, which is a multi-step enzyme-mediated biochemical process and the most abundant and complex post-translational modification in eukaryotic cells [17]. Additionally, glycoform distribution is a product of the interplay between the cell’s genome, transcriptome, and metabolome, as the subset of the glycan structures synthesised by a particular tissue or cell under different environmental conditions and physiological states is time-specific and dependent on the expression levels and activity of glycan-processing enzymes, as well as the availability of enzyme-related co-substrates, which is affected by metabolic function [18]. Although protein glycosylation takes place in both healthy and diseased tissues, glycoform distribution is considered to be disease-dependent [15]. It has been postulated that in autoimmunity, each autoimmune disorder could be characterised by a distinct glycan signature of immune cells and serum proteins [9]. These glycan signatures would be site-specific and quantified by the relative abundance of different glycan structures in these proteins [9].

The relative abundance of N-glycans decorating human plasma proteins has been found to remain surprisingly stable in healthy individuals, with notable variation majorly emerging due to aging, pathology, and/or lifestyle changes [19]. This, in addition to the ease of sample collection, makes human plasma glycoproteins excellent candidates for the discovery of biomarkers. Of particular interest are the N-glycans found on immunoglobulins, particularly immunoglobulin G (IgG), which is a glycoprotein which is abundantly present in human plasma/serum [20]. In general, immunoglobulins are the cornerstone of adaptive immunity, and mediate a number of effector immune responses, including antibody-dependent cellular phagocytosis (ADCP), antibody-dependent cellular cytotoxicity (ADCC), and complement-dependent cytotoxicity (CDC) [21]. Among their other roles—including securing protein solubility and conformation, as well as intracellular transport and clearance—immunoglobulin glycans are responsible for the regulation of these effector functions [21]. Interestingly, the presence of certain glycan structures in immunoglobulins has been associated with pro-inflammatory antibody activity [9,22], thus intensifying the need for a systematic study of immunoglobulin N-glycan profiles and their role in disease [23,24]. 

Significant developments in the elucidation of the role of glycosylation in cancer have been realised in previous decades [25,26,27,28], with the consensus that aberrant protein glycosylation, including that of IgG, is explicitly associated with known hallmarks of cancer [29,30]. The importance of this association is also reflected in the fact that the majority of FDA-approved diagnostic biomarkers for tumours used in clinical practice are glycoproteins or glycan-related [8,29,30]. Thus, N-glycan biomarker discovery for cancer diagnostics is a relatively mature field, with current research efforts largely being directed at the design of glycosylation-targeted immunotherapies [31,32,33,34,35]. In contrast, while some clinically-relevant protein-based biomarkers—including self-reactive IgG antibodies, also known as IgG autoantibodies—have been identified and used for the diagnosis of autoimmune diseases [36,37,38], the study of their glycosylation profiles is less developed. 

The aim of this article is to review human N-glycomics studies in the context of autoimmune diseases. Particular emphasis is placed on the identification of specific glycosylation traits for each autoimmune disease among the reviewed studies based on altered N-glycan profiles between healthy individuals and patients, and the presentation of their correlation with other relevant clinical parameters. Owing to the increased complexity of N-glycosylation, glycoinformatics approaches, including artificial intelligence and systems glycobiology, are also discussed due to their potential to be applied in N-glycan biomarker research as a means of enabling patient stratification through the interpretation of clinical data, as well as providing insight into the link between disease-specific N-linked glycosylation aberrations and their underlying molecular mechanisms, thus expediting the translation of such biomarkers into clinical practice. 

## 2. IgG Structure and N-Linked Glycosylation in Healthy Individuals

IgG is one of the five immunoglobulin isotypes in vertebrates (the others being IgA, IgD, IgE, and IgM), and is the most abundant in terms of its concentration in human serum, accounting for approximately 10–20% of the total plasma proteome [39]. Similarly to the other immunoglobulin isotypes, the structure of IgG is characterized by four polypeptide chains, covalently bound by disulphide bridges, consisting of two identical γ heavy (H) chains and two identical κ or λ light (L) chains. Each heavy chain is compartmentalised into four domains—one variable domain (VH) and three constant domains (C_H1_, C_H2_, C_H3_)—while each light chain comprises just two domains: one variable (V_L_) and one constant (C_L_). A further structural subdivision of IgG that is important for its function includes the Fab, representing the antigen-binding fragment of the IgG which is formed by the V_H_ and C_H1_ domains, and the Fc, representing the crystallizable fragment of the IgG which is responsible for the modulation of its effector functions through the binding to dedicated Fc receptors (FcγRs) [40,41]. With regard to its glycosylation, IgG has a conserved glycan at the N297 position (i.e., Asn-297) in each of its C_H2_ domains of the Fc region. This glycan typically corresponds to a core structure consisting of N-acetylglucosamine (GlcNAc) and mannose residues (Man) that can be further embellished with fucose (Fuc), galactose (Gal), sialic acid (NeuAc), and bisecting GlcNAc (Figure 1), with large population studies indicating that 30 main N-glycan structures are found in human serum IgG [42]. In healthy adults, the total serum IgG Fc is highly fucosylated (>90%), contains 35% agalactosylated (IgG-G0), 35% monoglycosylated (IgG-G1), 15% digalactosylated (IgG-G2), and 10–15% mono- and disialylated structures (IgG-S) [23]. Approximately 10% of circulating IgG also contains bisecting Fc glycans [23]. In contrast to IgG Fc N-linked glycosylation, there is no conserved N-linked glycosylation position in IgG Fab; rather, N-glycosylation sites can emerge during the somatic hypermutation of the variable domain [43]. In fact, it has been estimated that approximately 15–25% of serum IgG contains Fab N-glycans [44], which—compared to Fc N-glycans—are typically biantennary complex-type structures with a significantly higher extent of sialylation [45].

## 3. Differential Serum IgG N-Linked Glycosylation in Autoimmune Diseases

The presence of autoantigen-specific antibodies of the IgG isotype in human serum is considered a hallmark of various autoimmune diseases. However, the levels of serum autoantibodies alone do not seem to be associated with autoimmunity, as natural IgG autoantibodies can be found in abundance in serum from healthy individuals deprived of a potential antigen [46,47]. Their glycosylation, as quantified by different traits including galactosylation, sialylation, fucosylation, and bisecting GlcNAc, is likely a critical aspect of their role in the pathophysiology of autoimmune diseases. Indeed, experimental evidence corroborates the importance of glycosylation in the function of IgG autoantibodies, as their enzymatic deglycosylation leads to a loss of function in vivo [48]. Glycosylation, and particularly the extent of sialylation, confers a similar functional role in intravenous immunoglobulin (IVIg) preparations, which are IgG antibodies collected from donor sera, and are a common treatment option in autoimmune diseases [49]. In particular, the removal of terminal sialic acid moieties from IVIg abrogates its anti-inflammatory activity [50,51]. Furthermore, the link between glycosylation and autoimmunity may be promising as a therapeutic target, as the treatment with IVIg, which leads to reduced disease severity in patients with a neurological autoimmune disease, is also accompanied by a normalisation of specific glycosylation traits to the respective levels in healthy individuals [52]. Thus, the identification of differences in the N-linked glycosylation of total serum IgG, as well as autoantigen-specific antibodies, between healthy individuals and patients would not only enable accurate diagnosis but also guide the rational design of appropriate therapies. 

In order to facilitate the analysis presented herein, autoimmune diseases are grouped into two categories: rheumatic autoimmune diseases and autoimmune diseases with other pathophysiological features. Rheumatic autoimmune diseases are grouped according to the classification provided by the Harvard Medical School (https://www.health.harvard.edu/diseases-and-conditions/whats-the-deal-with-autoimmune-disease (accessed on 17 February 2022)). The altered glycosylation traits reported for each autoimmune disease in different studies comparing the serum IgG N-glycan profile between healthy individuals and patients are summarised in Table 1.

### 3.1. Rheumatic Autoimmune Diseases

The largest number of differential studies of IgG N-linked glycosylation in biomarker research published in the last 40 years is focused on rheumatic autoimmune diseases and, in particular, on rheumatoid arthritis [53,54,55,56,57,58,59,61,62,63,64,65,66,67,68,69,70,71,72,74,75,77,79,80,82,88,121,122,123], systemic lupus erythematosus [56,59,62,64,73,74,91,92,93,124], and, to a lesser extent, on Sjogren syndrome [59,63,73,74] and ANCA-associated systemic vasculitis [97,98,99,100,101,102,125].

#### 3.1.1. Rheumatoid Arthritis

Rheumatoid arthritis (RA) is the most common rheumatic autoimmune disease, and is characterised by systemic synovial inflammation, leading to joint destruction and bone erosion [126]. The impact of RA on patients is not limited to articular joint damage, and additional clinical manifestations include chronic inflammation affecting other organs, such as the blood vessels, eyes, skin, lungs, and heart [127].

The first study identifying the presence of aberrant IgG N-linked glycosylation between healthy individuals and patients with RA was carried out by Parekh et al. [82]. After evaluating approximately 1400 oligosaccharide structures from 46 IgG samples, this seminal work demonstrated that RA patients had significantly reduced galactosylation compared to healthy individuals, resulting in an increased relative abundance of complex biantennary N-glycans with terminal GlcNAc residues in one or both antennae. This work was not only a milestone in the identification of altered glycosylation traits in autoimmune diseases but was also one of the first demonstrations of glycan-based diagnostics. Subsequent studies have corroborated the prevalence of agalactosylated IgG glycoforms in RA patients, making reduced IgG N-linked galactosylation a well-established feature of this disease [53,54,55,56,57,58,59,60,61,62,63,64,65,66,67,68,69,70,71,72,73,74,75,76,77,78,79,80]. In addition to aberrant galactosylation, additional RA traits include reduced sialylation [55,68,72,78,81,82] and increased fucosylation [75,78,83]. The majority of these studies were carried out in the last 10 years, indicating that the concomitant advancement in analytical techniques has enabled the discernment of additional N-glycan features that can be used in tandem with aberrant galactosylation for biomarker discovery. An appreciable subset of the reviewed studies focused on the identification of correlations between IgG galactosylation levels and various biological and clinical parameters. It is important to note that because N-linked IgG galactosylation levels are reported to be age-related in healthy individuals [128], sample analysis results must first be age-corrected in order to enable statistically valid comparisons [53,65]. Using linear regression analysis for results from a 38-patient cohort, Tomana et al. [64] demonstrated that IgG galactose content and patient age were negatively correlated; however, this finding was not replicated by the same authors in a study 6 years later [66], with the authors pointing to the relatively small cohort size (i.e., 11 patients) as a plausible explanation. The patient cohort size may, nevertheless, not be the deciding factor in this instance because, in a study accounting for a larger number of patients (i.e., 50), Gindzienska-Sieskiewicz et al. [77] reported no correlation between IgG galactose content and patient age. Although earlier studies found no correlation between IgG galactose content and several clinical parameters, such as sex, race, the volume of packed red blood cells, radiographic grade, disability index, extra-articular manifestations, erosions, corticosteroid use, and RA-specific autoantibody levels [62,64], Bodman-Smith et al. found a high degree of correlation between IgG galactosylation in RA patients and their probable outcome [70]. In particular, the probable clinical outcome of two cohorts (A: 40 patients and B: 24 patients) was successfully predicted (with accuracies of 95% and 78%, respectively) through discriminant functional analysis based on agalactosylated IgG levels at the onset of the disease, combined with patient age, sex and grip strength. These findings highlight the importance of incorporating multiple clinical and/or serological parameters in the statistical analysis of such studies in order to draw clinically relevant conclusions.

A different approach to the interpretation of aberrant IgG galactosylation was followed by Axford et al. [69], whose study aimed to investigate the relationship between the extent of IgG galactosylation in RA patients with the activity of lymphocytic galactosyltransferase (GalT), which is responsible for the catalysis of galactose transfer from the UDP-galactose donor to an GlcNAc acceptor within the Golgi. Interestingly, GalT activity was found to have a negative linear correlation with agalactosylated IgG levels. Based on this finding, Bodman et al. [58] aimed to distinguish whether the reduced levels of IgG galactosylation were explicitly associated with the reduced GalT activity, or whether this galactosylation deficiency was caused by IgG modification by hydrolytic enzymes after secretion. Using IgGs produced by B-cells in vitro as a reference, it was shown that the latter hypothesis was probably not the case [58].

Among the different correlations discussed here, altered galactosylation may also be useful in facilitating patient stratification, as galactosylated IgG levels have been shown to be negatively correlated with disease activity in RA [53,65,77,79]. A similar negative correlation appears to exist for sialylation as well [78]. Importantly, this negative correlation for both glycosylation traits was particularly evident when monitoring galactosylation and sialylation levels in RA patients in order to assess their response to therapy. After treatment with various pharmacological agents, including monoclonal antibodies, it was found that RA improvement coincided with an increase in the galactosylation [78,129,130] and sialylation [129] levels. Finally, the prognostic potential of aberrant galactosylation as a biomarker in RA has also been investigated [55,79,131]. In a 10-year follow-up study considering large population cohorts, Gudelj et al. [55] observed that alterations in IgG galactosylation preceded the onset of RA-related symptoms by a median time of 4.31 years, which corresponded to the time that had elapsed between blood sampling at the beginning of the study and the time that a part of the cohort was diagnosed with RA within the follow-up period. Interestingly though, the authors reported that there appeared to be no significant statistical correlation between IgG galactosylation and the time preceding symptom manifestation, suggesting that aberrant galactosylation could be a pre-existing risk factor in RA.

In RA, the majority of disease incidence is represented by women, with a female to male ratio of 3:1 [132]. This imbalance is also reflected in the sex ratio of the patient cohort that was investigated in most of the studies reviewed herein [53,54,55,64,65,69,70,71,74,75,79,121]. Generally, within female RA patients, a large percentage (approximately 75–90%) shows an improvement in disease activity during pregnancy and a subsequent exacerbation, referred to as ‘flare’, after delivery [133]. Despite this observation, relatively few research efforts have focused on providing more insight into this phenomenon through IgG N-linked glycosylation studies of pregnant RA patients [57,68,72,123]. Earlier studies demonstrated that this pregnancy-induced amelioration of RA disease activity was associated with an increase in IgG galactosylation [57,123]. Despite this important finding, which is consistent with the independent observations mentioned above reporting a negative correlation between galactosylated IgG levels and disease activity, the patient cohort size in both studies was relatively small (i.e., 7 and 23, respectively) [57,123]. Additionally, no information was provided regarding disease exacerbation taking place postpartum. The comprehensive study of van de Geijn et al. [72], based on a much larger cohort of 148 patients, accounted for galactosylation levels of IgG1 and IgG2 both during pregnancy and postpartum, and corroborated the incidence of increased galactosylation during the first, which reached a maximum for both IgG1 and IgG2 during the third trimester. The subsequent reduction of galactosylation levels in the postpartum period could potentially explain the flare experienced by RA patients. A similar trend to that of galactosylation was observed for sialylation as well, which could be expected because these glycosylation traits are associated [72]. In a subsequent highly detailed study from the same group, Bondt et al. [68] investigated several glycosylation traits in pregnant RA patients, particularly galactosylation, fucosylation, sialylation, and bisecting GlcNAc in the four IgG subclasses. This work accounted for the largest patient cohort found in pregnancy-focused RA studies, with 219 RA participants. The maximum value for both galactosylation and sialylation was also found in the third trimester of RA patients for all IgG subclasses while, for bisecting GlcNAc and fucosylation, a minimum was generally observed for all IgG subclasses in the second and third trimesters, respectively. Importantly, this study underlined that the increased disease activity in RA was associated with reduced galactosylation independently of sialylation, thus providing more insight into the causative role of individual IgG glycosylation traits in the pathophysiology of RA.

Important developments in N-glycan biomarker discovery in RA have also been made while studying alterations in the glycosylation patterns of RA patients with respect to autoantigen-specific antibodies, particularly anti-citrullinated protein autoantibodies (ACPA). The N-linked glycosylation of ACPA has also been shown to be characterised by reduced galactosylation and sialylation compared to total IgG [84,85,86]. More recently, ACPA Fab N-glycans have been implicated in the pathophysiology of RA [134,135,136]. Rombouts et al. [134] showed that the majority of ACPA-IgG are glycosylated in their variable domains and, in a follow-up study, that these glycans are highly sialylated [135]. In their latest publication, Kissel et al. [136] aimed to elucidate the functional roles of glycans found in the variable domain of ACPA, suggesting that these glycans may mediate the activation of autoreactive B cells, and thus are, at least in part, involved in the dysregulation of the adaptive immune response in RA. These important findings pave the way for future investigations regarding the functional role of autoantigen-specific antibodies in rheumatic autoimmune diseases, further highlighting the potential of N-glycan biomarker discovery, combined with its diagnostic and prognostic value, in shedding light on the etiology of autoimmune diseases.

#### 3.1.2. Systemic Lupus Erythematosus and Sjogren’s Syndrome

Systemic lupus erythematosus (SLE) is a chronic rheumatic autoimmune disease that affects multiple organs, including the joints, skin, central nervous system, and kidneys [137]. Clinical manifestations can vary significantly, and high incidence rates are found in people of non-Caucasian ethnicities and women of childbearing age, with a female to male ratio of up to 13:1 [138,139]. Sjogren’s syndrome (SS) is also a chronic rheumatic autoimmune disease that is commonly manifested with mouth and eye dryness owing to the inflammation of the salivary and lacrimal glands [140]. This disease is associated with other rheumatic autoimmune diseases, including RA and SLE, but can also manifest itself alone (i.e., primary SS).

Compared to RA, the number of comparative serum IgG N-glycomics studies on SLE and SS is considerably smaller and, especially in earlier years, these studies were a subset of larger studies that focused primarily on RA [56,59,62,64,73,74,91]. Similarly to RA, reduced galactosylation has also been generally reported in patients with SLE [56,64,73,91,92] and SS [63,73]. Contrary to this finding, in an early comparative study to discern disease-specific aberration in IgG glycosylation in several autoimmune diseases, Parekh et al. [59] found that serum IgG was normally galactosylated in patients with SLE and primary SS, excluding, however, a seropositive patient subgroup showing higher levels of agalactosylated IgG. Interestingly, this subgroup met the diagnostic criteria for both SLE and primary SS [59]. A follow-up study from the same group a few years later [56] also reported reduced galactosylation in patients with SLE complicated by SS, reinforcing the notion that stratifying patients between closely-related diseases solely based on aberrant IgG galactosylation is likely impossible. The most comprehensive study regarding serum IgG N-linked glycosylation aberration in SLE patients was carried out by Vuckovic et al. [92]. This study was based on a large cohort comprising a total of 475 SLE patients, and was not limited to the identification of changes in IgG galactosylation, but extended to sialylation, fucosylation, and bisecting GlcNAc. While corroborating the findings of earlier studies with regard to reduced galactosylation, Vuckovic et al. [92] also reported reduced sialylation and increased bisecting GlcNAc. The general trend of downregulated galactosylation, sialylation combined with increased bisecting GlcNAc is also discerned in RA patients (Table 1); however, contrary to RA, it was shown in this particular study that, in SLE patients, fucosylation was notably reduced. This development shows promise for finding disease-specific biomarkers within rheumatic autoimmune diseases. An additional contribution of this work by Vuckovic et al. [92] is the indication that altered IgG glycans are associated with disease status, disease risk, and symptom severity in SLE patients, thus potentially opening up new avenues for the exploration of personalised treatments predicated upon aberrant IgG glycosylation. Regarding SS, fewer developments outside of downregulated IgG galactosylation have been identified, and the pertinent studies are more than 20 years old [63,73,96]. Due to the manifestation of SS in patients with RA and SLE, additional studies employing the improved analytical technologies available today would be beneficial for the discovery of more disease-specific IgG glycosylation traits in patients with SS in order to successfully predict and/or differentiate patients from other rheumatic autoimmune diseases through altered IgG glycosylation.

#### 3.1.3. ANCA-Associated Vasculitis

Antineutrophil cytoplasmic antibody (ANCA)-associated vasculitis (AAV) comprises a group of rheumatic autoimmune diseases, including polyangiitis (GPA), microscopic polyangiitis (MPA), and eosinophilic granulomatosis with polyangiitis (EGPA) [141]. AAV is characterised by the inflammation and eventual necrosis of blood vessels with heterogeneous clinical manifestations [141,142]. As suggested by the name given to this collection of diseases, circulating ANCA IgG is considered central to its pathogenesis, primarily targeting two autoantigens: proteinase-3 (PR3) and myeloperoxidase (MPO) [143,144].

Similarly to the other rheumatic autoimmune diseases presented here, early studies indicated that serum IgGs from ANCA-positive AAV patients show reduced galactosylation compared to healthy controls [97,98]. Subsequent studies have extended this research to additional glycosylation traits, including sialylation, which also appears to be downregulated in ANCA-positive AAV patients [99,100,101]. Focusing on GPA, Wuhrer et al. [102] investigated glycosylation aberration with respect to IgG1 and IgG2 between healthy individuals and patients. Significantly reduced total IgG galactosylation was found for both subclasses in GPA patients compared to controls of the same age. The total sialylation of IgG1 and IgG2 was also reported to be correlated with galactosylation and thus reduced in patients with GPA. Surprisingly, the presence of bisecting GlcNAc residues in the total IgG in GPA patients was shown to be decreased compared to healthy controls [102]. This appears to be a deviation from the differential glycosylation traits reported for the other rheumatic autoimmune diseases presented herein, which may indicate that aberrant bisecting GlcNAc could be monitored for differentiation between GPA and other rheumatic diseases, such as RA, SLE, and SS. Finally, the glycosylation profiles corresponding to anti-PR3 specific IgG1 and total IgG1 in patients with GPA were found to be similar; however, the total galactosylation and sialylation of IgG1 were found to be weakly or not correlated with their anti-PR3 specific IgG1 counterparts, respectively.

Aberrant glycosylation profiles in patients with AAV have also been reported to correlate with disease activity. Lardinois et al. [99] observed that there was a negative correlation between total IgG1 galactosylation and disease activity, quantified using the Birmingham Vasculitis Activity Score (BVAS), in samples from PR3-ANCA patients. This correlation appears to be autoantigen-specific, as MPO-ANCA samples did not show a correlation. This finding could provide a basis for the differentiation of PR3-ANCA- and MPO-ANCA-associated AAV diseases. With respect to other glycosylation traits, such as sialylation, fucosylation, and bisecting GlcNAc, no statistically significant association was observed with respect to disease activity, regardless of the autoantigen patient sample. In this study, IgG galactosylation levels were also shown to be reliable indicators of active PR3-ANCA patients, and were distinguishable from remission and healthy individuals [99]. Regarding autoantigen-specific IgG glycosylation, Espy et al. [101] reported a negative correlation between anti-PR3 IgG sialylation and BVAS in patients with GPA. However, this finding was not corroborated by the subsequent study of Wuhrer et al. [102], which indicated that the bisecting GlcNAc of anti-PR3 IgG was the only glycosylation trait negatively correlated with BVAS. This discrepancy could not necessarily be attributed to the size of the patient cohort, as it was comparable in both studies; rather, it could be the result of the different analytical methods used.

### 3.2. Autoimmune Diseases with Non-Rheumatic Pathophysiology

#### 3.2.1. Inflammatory Bowel Disease

Inflammatory bowel disease (IBD) refers to a constellation of disorders related to chronic inflammation in the gastrointestinal tract, including Crohn’s disease (CD) and ulcerative colitis (UC), and is associated with several symptoms, including fatigue, abdominal pain, and chronic diarrhea [145,146].

Similarly to all of the autoimmune diseases reviewed herein, reduced IgG N-linked galactosylation in patients with IBD has also been widely reported [59,64,73,103,104,105,106,107,108,109]. Within these studies, the earliest ones—being significant in terms of the introduction of antibody glycomic analysis concerning IBD—were limited regarding the patient cohort size used, which did not exceed 60 patients [59,64,73,103,104]. The study by Trbojevic-Akmacic et al. [106] investigated a markedly larger cohort of patients, namely 287 CD and 507 UC patients, and corroborated the findings of previous studies on downregulated IgG galactosylation in both CD and UC. Furthermore, a significant decrease in sialylation was found exclusively in CD. In a subsequent study from the same consortium, Simurina et al. [109] extended their analysis to 1065 CD patients and 1009 UC patients, now investigating the aberration of IgG subclass-specific glycosylation. The findings from their previous study, namely the downregulation of galactosylation in both CD and UC patients compared to healthy controls, as well as the downregulation of sialylation seemingly exclusive to CD, were confirmed in this study as well. Interestingly, a subclass- and disease-specific aberration in fucosylation was also observed. In particular, IgG1 fucosylation was increased in CD patients, but IgG2/3 fucosylation was decreased in UC patients. This result could be promising as a biomarker, which, along with the disease-specific clinical manifestation of IBD, could potentially facilitate the stratification of patients with CD or UC.

Throughout the last 40 years, research efforts have been undertaken to elucidate the potential correlation between the changes in IgG glycosylation in IBD patients and other parameters of clinical importance, such as C-reactive protein (CRP) levels and disease activity, with ambiguous results [73,103,104,108]. As early as 1990, Dube et al. [103] reported that galactosylated IgG levels were negatively correlated with CRP levels in CD patients. This correlation was not statistically significant in UC patients. However, in a subsequent study, Shinzaki et al. [108] found no such correlation, either for CD patients or UC patients. Regarding disease activity, the early study by Go et al. [104] reported that IgG galactose deficiency was positively correlated with clinical activity in CD patients but not in UC patients. In particular, the molar ratio of mannose to galactose in serum IgG was used as the monitored glycosylation trait. However, in a subsequent study, Bond et al. [73] found no association between the relative abundance of galactosylated glycan structures and disease activity for either CD or UC patients. In turn, Shinzaki et al. [108] showed that the ratio between the agalactosylated and digalactosylated fraction of fucosylated glycans (i.e., G0F/G2F) could be a promising biomarker, as it was positively correlated with active disease both in CD and UC patients. The ambiguity in these results could be an impetus for further investigation, particularly with regard to the selected glycosylation traits used to identify potential associations with important clinical parameters.

#### 3.2.2. Autoimmune Thyroid Diseases

Autoimmune thyroid diseases (AITD) represent a group of organ-specific disorders that dysregulate the function of the thyroid gland, with the most frequent forms being Hashimoto’s thyroiditis (HT) and Graves’ disease (GD) [147,148]. AITDs are characterised by the production of autoantibodies targeted at three main autoantigens, namely thyroid peroxidase (TPO), thyroglobulin (Tg), and thyroid-stimulating hormone receptor (TSHR) [148]. The prevalence of anti-TPO and anti-Tg autoantibodies is a known hallmark in HT, while GD is characterised by the presence of anti-TSHR autoantibodies [147,148].

Studies regarding IgG N-glycome analysis in AITD patients are centered on the glycosylation of autoantigen-specific IgGs and, particularly, anti-Tgs [111,112,149]. Yuan et al. [112] studied 32 patients with HT, and found that sialylation and fucosylation were increased compared to healthy controls. More recently, Li et al. [111] investigated the subclass-specific glycosylation profiles of anti-Tg IgG in patients with HT and GD. With respect to anti-Tg IgG1, it was shown that sialylation and fucosylation were also increased in HT patients compared to healthy individuals, a finding which is in agreement with the study of Yuan et al. [112], which, however, did not account for subclass-specific glycosylation changes. Interestingly, increased IgG1 galactosylation was reported for HT patients. Regarding patients with GD, no aberration was found in anti-Tg IgG1 glycosylation. However, this was also the case for both HT and GD patients with respect to anti-Tg IgG4 glycosylation, for whom it was shown that no differences could be discerned between healthy controls and patients. These results suggest that autoantigen-specific IgG biomarkers might not be well-suited for GD diagnosis; however, more studies are required in order to draw any reliable conclusions. Nonetheless, when it comes to distinguishing between patients with HT and GD, Zhao et al. [149] showed that fucosylation was significantly decreased in the former.

Regarding total serum IgG N-glycomic studies, Martin et al. [110] conducted a large investigation incorporating three independent patient cohorts looking for correlations of total IgG glycosylation with AITD and autoantibody levels. Fucosylation was shown to be decreased in AITD patients, and negatively correlated with anti-TPO autoantibody levels. This finding highlights the fact that aberrant glycosylation compared to healthy controls can manifest itself differentially in total serum IgG and autoantigen-specific IgG, indicating potential differences in the underlying regulatory mechanisms of glycosylation.

#### 3.2.3. Neurological Autoimmune Diseases

While a large subset of autoimmune diseases involves disorders that affect the nervous system [150], limited developments have been made regarding serum IgG N-glycan-based biomarkers compared to the other autoimmune diseases reviewed in this article. Relevant studies have reported altered glycosylation in patients with multiple sclerosis (MS) [113,151], Guillain–Barre syndrome (GBS) [52], chronic inflammatory demyelinating polyneuropathy (CIDP) [114], and myasthenic syndromes, including myasthenia gravis (MG) [56,115] and Lambert-Eaton myasthenic syndrome (LEMS) [115].

MS is a chronic neuroinflammatory autoimmune disorder affecting the brain and the spinal cord. Its clinical manifestation is associated with lesions within the central nervous system (CNS) that promote demyelination and neurodegeneration, with the eventual disruption of neuronal signalling [152]. Wuhrer et al. [151] focused on changes in IgG1 glycosylation, as intrathecal IgG synthesis is a hallmark of MS, both in the CSF and serum of a cohort of 48 MS patients. Interestingly, compared to healthy controls, the glycosylation profiles obtained from serum-derived IgG1 were not significantly altered. However, in a more recent independent study with a larger patient cohort (i.e., 83 MS patients), serum IgG core fucosylation was found to be significantly reduced, while a higher prevalence of high mannose glycans was also observed [113]. Interestingly, antennary fucosylation in patients’ total plasma proteins was found to be upregulated, thus indicating the multi-faceted role of the same glycan structures in disease, further complicating the identification of disease-specific N-glycan changes that can be promptly translated into clinical practice.

GBS and CIDP are also neuroinflammatory demyelinating autoimmune disorders which, contrary to MS, involve the peripheral nervous system (PNS) [153,154]. Although rare, the incidence of GBS has recently been reported to be a very rare side-effect after vaccination with particular vaccines against COVID-19 [155,156]. Regarding IgG glycosylation studies, Fokkink et al. [52] investigated subclass-specific N-linked glycosylation patterns in 174 patients with GBS. In particular, the N-linked glycosylation changes in IgG1 and IgG2 were monitored in patients at the onset of GBS and after treatment with IVIg. Before the administration of IVIg, and compared to healthy individuals, galactosylation was downregulated in serum IgG1 and IgG2, while sialylation was only downregulated in serum IgG2. Interestingly, this pro-inflammatory state was shown to partially normalise in patients after treatment, which could potentially be attributed to elevated levels of galactosylation and sialylation in IVIg-contained IgG. These findings demonstrate the potential of monitoring these glycosylation traits in order to assess disease activity and the response to therapy in GBS. The study of IgG sialylation and its correlation with disease activity has also been a topic of interest in patients with CIDP. Wong et al. [114] reported that IgG sialylation was reduced in CIDP patients compared to healthy individuals, and found that IVIg treatment increased sialylation levels. This study suggested the use of IgG sialylation combined with the ratio of sialylated to agalactosylated IgG as measures to monitor disease activity and the response to therapy in CIDP.

Myasthenic syndromes, such as MG and LEMS, are neurological autoimmune disorders that involve defects in neuromuscular transmissions, leading to muscle weakness [157,158]. Similarly to MS, GBS, and CIDP, limited research efforts have focused on IgG N-glycan biomarker discovery in MG and LEMS. Notably, Selman et al. [115] studied the N-linked glycosylation profiles associated with serum IgG1 and IgG2 in patients with both MG and LEMS. Importantly, the reported glycosylation changes were found to be subclass- and disease-specific. With respect to galactosylation, patients with both diseases exhibited reduced levels; however, in MG this reduction appears to be related solely to serum IgG1. The sialylation of IgG1 and IgG2 did not differ in MG and LEMS patients compared to healthy individuals, with fucosylation following the same trend in MG. Nonetheless, in LEMS patients, downregulated fucosylation was found for serum IgG2. Finally, the bisecting GlcNAc level in both IgG subclasses was unchanged in patients with MG, while, contrastingly, in patients with LEMS, this trait was upregulated for serum IgG1 and IgG2.

It is important to note that the diagnosis of neurological autoimmune diseases, particularly those characterised by demyelination, is difficult due to the similarities in their clinical manifestation [159]. Thus, the discovery of disease-specific IgG N-glycan biomarkers would greatly benefit the handling of diseases with such pathophysiology. Despite existing research efforts, more studies in this direction are required in order to draw informative conclusions in this regard.

## 4. Integration of Systems Glycobiology and Artificial Intelligence Approaches in N-glycomics Biomarker Discovery in Autoimmune Diseases

The discovery of N-glycan biomarkers for autoimmune diseases based on differential serum IgG glycosylation is a laborious and time-consuming endeavor requiring multi-disciplinary collaborations. As shown in the studies reviewed in this article, despite the identification of notable differences in the N-linked glycosylation profiles of IgGs in various autoimmune diseases (Table 1), significant ambiguity is still present with regard to disease-specific changes, hampering the establishment of biomarkers that could be promptly translated into clinical practice. Furthermore, the complexity associated with the mechanisms of glycan biosynthesis and N-linked glycosylation regulation further complicates the explicit mapping of serum glycosylation profiles to aberrant cellular functions. Computational approaches based on systems glycobiology and artificial intelligence/machine learning could facilitate the incorporation of N-glycan-based biomarkers into diagnosis, prognosis, and monitoring (Figure 2). They could further suggest treatment avenues for autoimmune diseases by providing insight into the mechanistic relationships between biological parameters and the resulting IgG glycoforms, helping to infer glycan motifs that allow for patient stratification with respect to specific disease subtypes and/or disease progression.

### 4.1. Disease-Specific Glycomics Databases

The need to formalise the analysis of glycans and integrate the glycomic data produced from a multitude of research efforts into dedicated databases and repositories that would facilitate their visual interpretation and the extraction of meaningful biological information is by no means a new idea [160]. In order to this end, significant progress has been made in recent decades in integrating not only glycomic data but also other omic data in curated databases and repositories [161,162,163,164]. These resources are based on information for numerous glycan structures, glycoproteins, glycosylation-relevant genes, and even glycan–protein interactions [162]. In order to facilitate pertinent research efforts, centralised and integrative web portals—incorporating multiple bioinformatic resources relevant to glycobiology—have been developed recently and are available to users. These portals are GlyGen [165] in the United States (https://www.glygen.org/ (accessed on 17 February 2022)), Glycomics@ExPASy [166] in Europe (https://www.expasy.org/search/glycomics (accessed on 17 February 2022)), and GlyCosmos [167] in Japan (https://glycosmos.org/ (accessed on 17 February 2022)). The systematisation of such information is a decisive step towards establishing the study of glycans as an integral part of biology and immunology. However, distinguishing which pieces of information need to be extracted from various databases and repositories in order to gain insight into glycan-based biomarker research might be cumbersome, and would inevitably require a strong background in glycobiology. This would potentially hamper the immediate adoption of such biomarkers by clinicians. This issue is exacerbated by the minimal existence of disease-specific glycomic databases. An important exemption is GlyConnect (https://glyconnect.expasy.org/ (accessed on 17 February 2022)), which is a Glycomics@ExPASy integrated platform that allows the collection, monitoring, and visualisation of glycobiological data, with a particular focus on the characterisation of the molecular components associated with protein glycosylation [168]. Different data types are available in GlyConnect, and can be accessed under specific categories, one of which is ‘Diseases’. Information about glycan types (e.g., N-linked, O-linked, free), as reported in pertinent scientific references, can be found for several diseases regarding different glycoproteins of interest, with the user also being given the option to focus on and explore glycan types regarding human IgG alone. Currently, with regard to the types of disease, great attention is being paid to different types of cancer, with N-linked glycans associated with different autoimmune diseases, mainly related to RA, SLE, and SS. The expansion of GlyConnect to include more references on already-existing entries for autoimmune diseases (e.g., RA, SLE, and SS), and to gather more data on other autoimmune diseases, would be an important step in the systematisation of glycobiological data for biomarker discovery in autoimmune diseases, while at the same time expediting the clinical translation of this field by helping clinicians without a relevant glycobiological background develop a better overview of the existing knowledge.

### 4.2. Artificial Intelligence for Glycomic Analysis and N-glycan Biomarker Discovery

In recent years, artificial intelligence (AI) has found many applications in healthcare, with machine learning (ML) techniques being particularly useful in clinical research aiming to improve disease management and enable precision medicine [169,170]. ML can facilitate disease diagnosis, prognosis, and the stratification of patients with similar diseases and/or disease types by leveraging diverse datasets, including clinical imaging, electronic health records, and omic data [171,172]. Several AI and ML applications have focused on diagnosis, prognosis, the identification of disease subtypes, and even drug development with regard to autoimmune diseases, including RA, SLE, IBD, and MS [173,174]. However, the wealth of data types used in these applications glaringly misses a very important piece of the puzzle: glycomic data. Indeed, AI approaches integrating glycosylation-related information within the context of personalised medicine, and especially towards autoimmune disorders, are scarce. Most advancements in this regard can be found in ML-powered glycomic analysis, with artificial neural networks (ANN) and deep learning (DL) being applied to annotate glycan structures from data [175,176], and to predict glycosylation sites [177,178,179]. Cancer-related advancements have been realised as well, including the employment of kernel classifiers, such as support vector machines (SVM), to identify glycan biomarkers in leukemic cells [180,181,182,183]. Regarding autoimmune diseases, several studies have built logistic regression-based classifiers to infer associations between disease status and predictors such as age, sex, and IgG glycosylation traits in juvenile-onset RA, SLE, AAV, IBD, autoimmune cholestatic liver diseases (ACLD), and MS [88,92,99,106,109,113,125,184]. The models have had appreciable success in distinguishing patients from controls, and in the classification of patients based on disease severity, with their predictive power being highly improved by the inclusion of glycosylation traits as predictors. Data-driven predictive modelling, such as that demonstrated in these studies, has the potential to expedite the clinical adoption of N-glycan biomarkers by providing an assessment of the impact of differential glycosylation in autoimmune disease diagnostics based on quantitative metrics, such as sensitivity and specificity. These metrics provide a concrete foundation to evaluate false/true positive/negative rates that can be directly used by clinicians without any background in glycobiology.

### 4.3. Lessons Learned from Biotherapeutics Manufacturing: Systems Glycobiology Approaches

Systems glycobiology strives to develop, simulate, and analyse glycobiological systems at the molecular and cellular level through the integration of multiple omic data types [35,185]. In the last few decades, several research efforts have been undertaken to mathematically model N-linked protein glycosylation, with the aim to predict the glycosylation profiles of biotherapeutic products, such as monoclonal antibodies, and to provide insight into the glycosylation machinery itself [186]. Recently, such mathematical models, varying in complexity and modelling approaches, have had appreciable success in the prediction of glycoforms [187,188,189,190], as well as in the successful reconstruction of the secretory pathway [191] in Chinese Hamster Ovary (CHO) cell cultures. In these mathematical models, the prediction of the N-linked glycosylation profiles for a given protein by a specific cell line was based on a given glycosylation reaction network that associated the intracellular glycosylation mechanisms with the final N-glycan structures. Such glycosylation reaction networks are essential for the quantitative analysis of biological data in the field of glycomics. In this regard, the early work of Krambeck et al. [192] was a critical step towards the automated in silico construction of glycosylation reaction networks. Their model, referred to as “KB2005”, considered a very large network of 22,871 reactions that were catalysed by 11 enzymes, leading to the prediction of 7565 distinct glycan structures with variable extents of different glycosylation traits, such as galactosylation and fucosylation. By adjusting the concentrations of the enzymes involved in this particular glycosylation reaction network, the experimental distributions of the glycoforms of a recombinant protein in CHO cells were successfully predicted. The same group generalised KB2005 to a more flexible modelling platform, with the updated name “KB2009”, which incorporated 19 glycosylation enzymes and enzyme reaction rules to define enzyme specificities [193]. Based on these specificities and given enzyme concentrations and kinetic parameters, a glycosylation reaction network could be automatically generated, and could predict the relative abundances of N-linked glycans. More interestingly, and closely related to the discovery of N-glycan biomarkers, KB2009 was able to discern differences in enzyme activities between normal and malignant human monocytes through the analysis of pertinent N-glycan mass spectroscopic data. Through the integration of transcriptomic data related to enzyme expression levels, the KB2009 modelling platform was employed to infer enzyme activity changes using mass spectroscopic data from human prostate cancer cells [194]. KB2009 has also influenced the independent development of similar N-linked glycosylation modelling platforms, such as the K2014 platform [195]. Although such system glycobiology approaches can find immediate application in the identification of glycoengineering targets to improve the quality and efficacy of biotherapeutic products, as well as in inferring differences in the underlying glycosylation mechanisms among different cell lines [196], they could potentially benefit N-glycan biomarker research in autoimmune diseases by quantitatively analysing the differential N-linked glycosylation profiles between healthy individuals and patients. At least in principle, these approaches could integrate the data for the different IgG glycosylation traits which are most commonly monitored in patients with autoimmune disorders (i.e., galactosylation, sialylation, fucosylation, and bisecting GlcNAc); they aim to identify biological reasons for their aberration based on the N-linked glycosylation machinery of B lymphocytic cells. If this is the case, and if clinically-consistent conclusions can be drawn regarding B-cell-related defects in enzyme functions and/or differential enzyme expression levels and activity, then these defects could provide a mechanistic insight into the link between N-glycosylation and the pathophysiology of autoimmune disorders, and, ultimately, could point to potential therapeutic targets and guide the rational design of advanced therapeutics. Nevertheless, it is important to note that there is a caveat when it comes to the direct employment of systems glycobiology approaches for the in silico generation of glycosylation reaction networks using human serum IgG samples. This caveat arises from the fact that these approaches were developed on the basis of the integration of reaction rules and specificities of enzymes that are involved in the protein secretory pathway of eukaryotic cells. However, adding to the existing complexity of N-linked glycosylation, there is, albeit still ambiguous, the possibility that B-cell-independent post-secretory modifications of IgG N-glycans can occur by glycosyltransferases in serum [197,198]. If this is the case, and N-linked glycosylation is not limited to the secretory pathway in B cells, then significant adjustments to the existing glycobiological modelling platforms would need to be implemented in order to be able to discern biologically- and clinically-relevant information about the N-glycan aberration in patients with autoimmune disorders. Thus, for the time being, prospective users of such modelling platforms should tread cautiously.

## 5. Conclusions

Glycomic data have long been underrepresented in biomarker research compared to other types of omics data. However, the involvement and functional role of glycans in the pathophysiology of various diseases, including autoimmune disorders, has the potential to provide insight into their etiology and the development of tailored therapies. In the last 40 years, important findings have emerged with regard to the discovery of IgG N-linked biomarkers for autoimmune diseases, adding another promising tool to the clinical toolkit for diagnosis, prognostics, and identification of disease subtypes. Regarding the differential N-linked glycosylation profiles between healthy individuals and patients, the greatest advance was made for rheumatic autoimmune disorders, with limited studies concerning other important autoimmune diseases, including autoimmune thyroid diseases and neurological disorders. While a small number of studies regarding particular groups of autoimmune disorders and/or studies with a relatively small cohort of patients are indeed valuable in order to gain insight into disease-specific aberrations in glycosylation without the need for extensive analysis, they are not conducive to drawing definitive conclusions regarding potential glycosylation-related hallmarks in autoimmunity. Such conclusions would greatly benefit from longitudinal studies monitoring patients over extended periods of time in order to help autoimmunity research catch up with respective developments in cancer. An appreciable subset of the studies reviewed herein includes quantitative metrics, such as sensitivity and specificity. This has direct implications for the adoption of N-glycan biomarkers by clinicians; thus, it is recommended to include such quantitative criteria in subsequent studies in this field. In this regard, the quantitative analysis and understanding of biological data can be systematised through the use of glycoinformatics approaches enabling personalised medicine. It is envisaged that novel tools, namely data-driven predictive modelling and AI-powered glycomic analysis, as well as more mature systems glycobiology approaches could play a critical role in expediting the clinical translation of N-glycan biomarker discovery in autoimmunity.

## Figures and Tables

**Figure 1 ijms-23-05180-f001:**
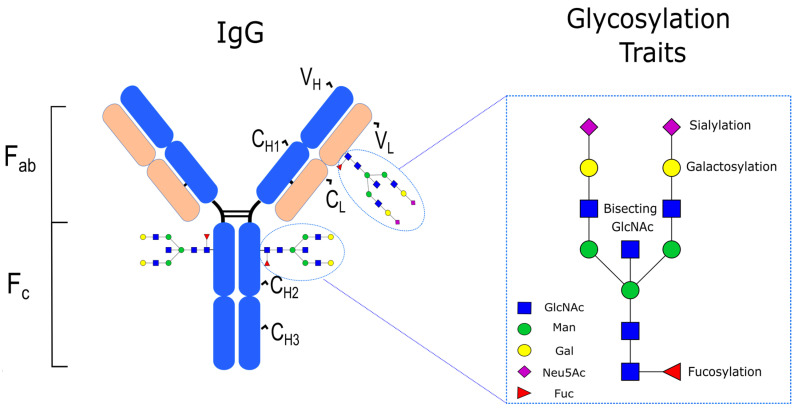
IgG structure and glycosylation traits.

**Figure 2 ijms-23-05180-f002:**
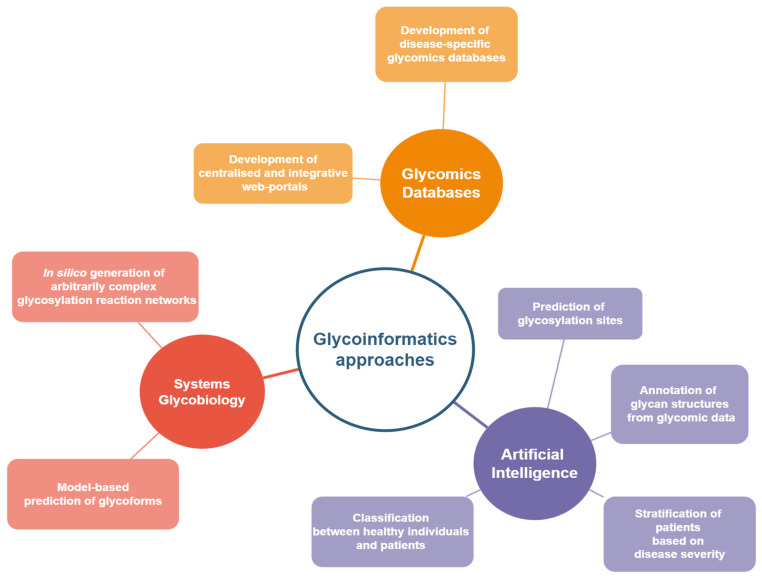
Current trends and developments in glycoinformatics within biomarker discovery.

**Table 1 ijms-23-05180-t001:** Comparative serum IgG N-linked glycosylation studies between healthy individuals and patients with autoimmune diseases.

Disease	Altered IgG Glycosylation Traits in Patients	
	Total IgG	Autoantigen-Specific Antibodies
Rheumatoid arthritis	Galactosylation ↓ [53,54,55,56,57,58,59,60,61,62,63,64,65,66,67,68,69,70,71,72,73,74,75,76,77,78,79,80] Sialylation ↓ [55,68,72,78,81,82] Fucosylation ↑ [75,78,83] Bisecting GlcNAc ↑ [63,68,73,76]	Galactosylation ↓ [75,79,84,85] Sialylation ↓ [86] Fucosylation ↑ [84]
Juvenile idiopathic arthritis	Galactosylation ↓ [53,60,62,63,73,74,87,88] Sialylation ↓ [88] Fucosylation ↑ [60] Bisecting GlcNAc ↑ [63,73]	
Osteoarthritis	Galactosylation ↓ [71,73,82] Sialylation ↓ [82]	
Spondyloarthropathies	Galactosylation ↓ [73,89,90]	
Systemic lupus erythematosus	Galactosylation ↓ [56,64,73,91,92] Sialylation ↓ [92,93] Fucosylation ↓ [92] Bisecting GlcNAc ↑ [92]	
Neonatal lupus	Galactosylation ↓ [94]	
Lupus nephritis	Galactosylation ↓ [95] Bisecting GlcNAc ↑ [95]	
Sjogren’s syndrome	Galactosylation ↓ [63,73] Sialylation ↓ [96] Bisecting GlcNAc ↑ [63]	
ANCA-associated systemic vasculitis		Galactosylation ↓ [97,98] Sialylation ↓ [99,100,101] Bisecting GlcNAc ↓ [102]
Crohn’s disease	Galactosylation ↓ [59,64,73,103,104,105,106,107,108,109] Sialylation ↓ [106] Fucosylation ↑ [109] Bisecting GlcNAc ↑ [73]	
Ulcerative colitis	Galactosylation ↓ [59,73,103,104,105,106,107,108,109] Fucosylation ↓ [109] Bisecting GlcNAc ↑ [73]	
Hashimoto’s thyroiditis	Fucosylation ↓ [110]	Galactosylation ↑ [111] Sialylation ↑ [111,112] Fucosylation ↑ [111,112]
Multiple sclerosis	Fucosylation ↓ [113]	
Guillain-Barre syndrome	Galactosylation ↓ [52] Sialylation ↓ [52]	
Chronic inflammatory demyelinating polyneuropathy	Galactosylation ↓ [114] Sialylation ↓ [114]	
Myasthenia gravis	Galactosylation ↓ [115]	
Lambert-Eaton myasthenic syndrome	Galactosylation ↓ [115] Fucosylation ↓ [115] Bisecting GlcNAc ↑ [115]	
Coeliac disease	Galactosylation ↓ [116]	
Type 1 diabetes	Galactosylation ↓ [117]	
Myositis	Galactosylation ↓ [118]	
Autoimmune hemolytic anemia	Galactosylation ↓ [119]	Galactosylation ↓ [119] Sialylation ↑ [119] Bisecting GlcNAc ↓ [119]
Antiphospholipid syndrome		Sialylation ↓ [120]

↓ = decreased; ↑ = increased.

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
