# Peer review of "Immunoglobulin G N-glycan Biomarkers for Autoimmune Diseases: Current State and a Glycoinformatics Perspective"

_ijms, 2022, doi:10.3390/ijms23095180_

Round 1

Reviewer 1 Report

This review comprehensively covers the field of glycan biomarkers for autoimmune diseases. The complexity of both glycosylation and autoimmune disorders makes the identification of biomarkers very difficult, and the authors successfully outlined the challenges, while offering valuable perspectives on how to move the field forward. In addition, useful references to databases are provided. One addition that would be very helpful would be to better discuss the current limitations in differential diagnosis between similar autoimmune disorders on the basis of autoantibody specificity. The manuscript dispenses with this subject very quickly in lines 135-138, and the reader is left to wonder what the relationship is between standard diagnosis and disease evaluation methods vs. glycosylation monitoring. The manuscript is overall well-organized and generally well-written, but it is wordy. As the volume of literature reviewed is so great, any effort by the authors to lighten the text would be appreciated by readers. For example, the sentence:

“To this end, this study provides a critical overview of the existing literature regarding human IgG N-glycomics studies with focus on identifying disease-specific alterations observed for different glycosylation traits.”

could be reduced to:

“This study provides a critical overview of the literature on human IgG N-glycomics with focus on identifying disease-specific glycan alterations.”

There are many other places where the text could be shortened without losing information or clarity.

The following is a list of minor comments:

Line 28: Why are cells and tissues given as examples of autoantigens? The autoantigens are presumably molecules.

Compound sentences take a comma before the “and” (lines 32-34 and other places in the text): The etiology of these diseases, characterised by both genetic predisposition and environmental triggers, has yet to be completely elucidated, and effective treatment hinges on timely diagnosis and monitoring [1,3,4].”

Line 81: The correct wording is “In contrast,” rather than “On the contrary,” since the immaturity of glycan analyses in autoimmunity does not contradict the maturity of the cancer field but rather contrasts with it.

Line 92: “less developed” seems more appropriate than “less pronounced”

Lines 127-129: This sentence is ambiguous, since, grammatically, “somatic hypermutation” could refer to the introduction of N-glycosylation sites rather than the general introduction of mutations into the variable domain. A better option would be: “N-glycosylation sites can emerge during somatic hypermutation of the variable domain [43].”

Lines 130-131: The text says that the Fab and Fc glycosylation are typically different, but the figure shows them to be identical.

Line 137: the term “natural IgG autoantibodies” should be explained.

Line 154: What is meant by “designate” the rational design of?

Line 174: It should probably be “is not limited to articular joint damage, and” rather than “but”

Lines 88-91: This sentence is unnecessary.

Lines 227-229: This sentence is a bit vague. Since this is a new paragraph, it is not 100% clear that “disease” still refers to RA. If it does, why is it surprising that the correlation is robust to how disease is quantified? Presumably the various methods of measuring disease correlate well with one another.

Line 229: should “negative” be added before “correlation” here as well, since otherwise the next sentence (“this negative correlation for both”) is confusing.

Lines 239-241: This sentence and the one before are a bit confusing. Was the sampling before disease diagnosis or the other way around? How could it be that people were sampled for this study before they acquired RA symptoms and were diagnosed? Is the “time elapsed” instead meant to refer to disease flare-ups? In any case, it is unclear why the conclusion is drawn from this study that “aberrant galactosylation is explicitly involved in the pathology of RA.” The suggestion that it could be a pre-existing risk factor is more logical.

Line 242: There should be a comma after “women”

Lines 247-248: Not clear what information this sentence provides.

Line 344: It should be just “indicated” and not “have indicated”

Line 345-346: would “Subsequent studies have extended this research to” be better phrasing?

Lines 442-443: the use of “however” twice in a row is awkward

Line 509: Should be: “It is important to note that the diagnosis”

Line 540: “unequivocally” is unnecessary, as “decisive step” makes the point strongly enough.

Lines 553-554: Regarding, “with a particular focus on facilitating the characterisation of the molecular components associated with protein glycosylation,” the broad generality of everything following “facilitating” contrasts with “focus.” Where is the focus?

Line 598: “quite literally translate”? In what way is it literal? Is this last sentence of the paragraph necessary?

Line 611: what does “in principle” add?

Line 671: would “the greatest advance has been made for rheumatic autoimmune disorders” be better phrasing?

Line 682: should it be “biomarkers by clinicians”?

Reviewer 2 Report

This review is well organized and well written. I have some comments.

1. The name of the disease 'juvenile-onset rheumatoid arthritis' is old. How about using 'juvenile idiopathic arthritis (juvenile onset rheumatoid arthritis or juvenile chronic arthritis)'?

2. Autoantigen-specific antibodies have also been reported in juvenile idiopathic arthritis and juvenile onset Sjögren's syndrome. Please refer to the following paper.

Maeno N, Takei S, Fujikawa S, Yamada Y, Imanaka H, Hokonohara M, Kawano Y, Oda H. Antiagalactosyl IgG antibodies in juvenile idiopathic arthritis, juvenile onset Sjögren's syndrome, and healthy children. J Rheumatol. 2004 Jun; 31 (6) ): 1211-7.

3. I think it would be better if there was another effective chart. 
